# Changes of perceptions and behaviours during the phases of COVID-19 pandemic in German elderly people with neurological disorders: an observational study using telephone interviews

Hannah M Zipprich,[1] Aline Schönenberg,[1] Ulrike Teschner,[1] Tino Prell  [1,2]

¹Department of Neurology, Jena University Hospital, Jena, Thüringen, Germany
²Center for Healthy Aging, Jena University Hospital, Jena, Thüringen, Germany

**Correspondence to**
Dr Tino Prell;
tino.prell@med.uni-jena.de

## ABSTRACT

**Objectives** Describing perceived limitations in everyday life, psychological burden and approval to easing of measures during the COVID-19 phases in elderly people with neurological disorders.

**Design** Observational, prospective study

**Setting** This is a monocentric study conducted at a university hospital in Germany.

**Participants** Overall, 452 elderly people participated in the NeuroGerAdh study (DRKS00016774) and were interviewed by telephone between 18 March and 30 August 2020.

**Results** Overall, 307 (67.9%) patients had relevant limitations in daily life due to the measures. These limitations significantly decreased during the pandemic phases. At the beginning of the pandemic, people complained about restricted social contacts and mobility, which were the most common reasons for perceived limitations in daily life. Later, since June 2020, wearing a mouth–nose mask had become the main reason for perceived limitations. In the elastic net regularisation, model higher perceived limitations in daily life were among others associated with younger age and earlier pandemic phases. Higher psychological burden was mainly associated with early pandemic phase, younger age and depression. The perceived psychological burden decreased as the pandemic phases passed, even though the reasons for psychological burden (anxiety or fear of infection, insecurity and concerns) did not remarkably change during the phases. From 16 June 2020, the patients were asked whether they approve the easing of measures. Sixty-seven of 136 patients (49.3%) approved and 55 (40.4%) did not. The common reasons for disapproval were fear of increased risk of infection and irresponsible behaviour of other people.

**Conclusion** While limitations in daily life decreased during the study period, anxiety remains a common psychological burden in elderly sick people, and this needs special attention. Accordingly, most people do not approve easing of measures. Special strategies are needed to cope with changing measures during the COVID-19 pandemic.

### Strengths and limitations of this study

► The inclusion of under-represented elderly sick people.
► The availability of clinical data for depression, cognitive ability and mobility for the telephone-interviewed participants.
► The monocentric study design and focus on people with neurological disorders.
► The cross-sectional design does not allow statements about causality.

## INTRODUCTION

SARS-CoV-2, a novel virus causing COVID-19 infection, has resulted in a pandemic. Local and national governments have taken unprecedented measures in response to the outbreak of SARS-CoV-2-induced COVID-19, including the isolation of patients, enforcement of quarantine of all contacts, cancellation of public transportation, exit controls, travel restrictions, social contact restrictions and requirement of people to wear mouth–nose masks.[1 2]

Considering the globally and locally rising infection rates, the German federal state issued different ordinances to contain the spread of the virus. The first ordinance published in March 2020 included a lockdown with limited social contacts, maintaining a minimum distance of at least 1.5 m in public and closing of service establishments and restaurants as well as teaching facilities. A maximum of two people from different households were allowed to meet (see also Thüringer Verordnung über erforderliche Maßnahmen zur Eindämmung der Ausbreitung des Coronavirus SARS CoV 2, 2020). Since April 2020, governments have been requiring people to

wear mouth–nose masks in shops and public transportation. In May 2020, Germany announced relevant easing of measures against the coronavirus pandemic: resuming regular operating hours of kindergartens and schools, reopening of restaurants and allowing meetings of up to five people from different households. Since the beginning of June 2020, further easing of measures has been occurring. For example, visits to nursing homes and larger family celebrations are possible again. Thuringia, in particular, was the first federal state to lift the contact restrictions. The new standard regulations deployed since June 2020 still require people to practice social distancing and wear mouth–nose masks in shops and public transportation, but contain only recommendations and no rules for restricting social contacts (see also Thüringer Verordnung zur Neuordnung der erforderlichen Maßnahmen zur Eindämmung der Ausbreitung des Coronavirus SARS-CoV-2, 2020).

It is understandable that such drastic measures have a considerable impact on the physical and psychological well-being of the population.[3–5] In particular, higher levels of depression, stress, and anxiety were found during COVID-19 pandemic.[5–7] Especially, the elderly population, being particularly vulnerable to severe cases of COVID-19, is at risk of increased anxiety and depression.[8] So far, however, the effect of the measures in the older population and changes in perceptions when measures were being relaxed have not been sufficiently investigated. Furthermore, little is known about how the measures are perceived by older people with chronic diseases or relevant functional impairments. In our opinion, the results of surveys of younger or community-dwelling elderly people, for example, COVID-19 Snapshot Monitoring (COSMO),[9] are not necessarily transferable to this cohort. Therefore, we interviewed elderly people who were previously hospitalised in our hospital because of a neurological disorder. We were particularly interested in how the perception of the measures changes over time. Therefore, we explored the impact of the measures during the three phases of the coronavirus pandemic in Germany: phase 1 (18 March 2020–15 May 2020) with the lockdown, phase 2 (16 May 2020–5 June 2020) with the first easing of measures and phase 3 (since 6 June 2020) with further easing of measures.

## METHODS
### Study design and assessments
The people who were enrolled in the NeuroGerAdh study (DRKS00016774) were interviewed via telephone between 18 March 2020 and 30 August 2020. The NeuroGerAdh study is a longitudinal observational study of predictors of non-adherence to medication in patients with neurological disorders.[10] Here, elderly people who were treated at the Department of Neurology at Jena University Hospital were consecutively recruited and assessed using comprehensive geriatric assessment and an adherence questionnaire and two follow-up telephone calls to explore changes in medication after discharge from hospital. All patients provided written informed consent. During the COVID-19 pandemic, the following questions were added to the interview:

1. Are or were you infected with coronavirus? Yes/no (if yes, when?)
2. Are or have you already been quarantined? Yes/no (if yes, how long?)
3. For which activities do you still leave the house/apartment?
   a. Shopping.
   b. Visiting to the doctor.
   c. Walking/hiking/cycling.
   d. Working in the garden.
   e. Doing sports.
   f. Visiting family.
   g. Meeting with friends.
   h. Not at all, I am being cared for.
4. Please indicate on a scale of 0–10 how much you are limited in daily life by the measures against COVID-19 (0 being 'no limitation of my daily life' and 10 being 'strongest restrictions of my daily life'). If >0, please specify what you are limited by/what you feel is a limitation.
5. Please indicate on a scale of 0–10 how much of a psychological burden the restrictions put on you (0 being 'not burdening at all' and 10 being 'very strongly burdening').
6. Since 16 June 202,0 we additionally asked, 'Do you approve the easing of the measures?' Yes/no (if no, why?)
7. From 16June 2020, the people were also asked, 'Do you approve the easing of the measures?' Yes/no (if no, why?)

Answers to questions 4–7 were evaluated qualitatively and categorised into specific subject areas. In addition, the following data were used for analyses: age, gender, main neurological disorder, marital status (single, divorced/widowed or married), level of education (high: German Abitur or university; medium: German Realschule or the General Certificate of Secondary Education; low: German Hauptschule or no school), cognitive function (Montreal Cognitive Assessment (MoCA)), depressive mood (Beck Depression Inventory II (BDI-II)), patient–physician relationship (Health Care Climate Questionnaire (HCCQ)) and Timed 'Up and Go' (TUG) test in seconds to assess a person's mobility and risk of falling.

### Patient and public involvement statement
Patients or the public were not involved in the design, or conduct, or reporting, or dissemination plans of our research.

### Statistical analysis
Statistical analysis was performed using SPSS (V.25.0; IBM) and R V.3.6.2 (R Foundation for Statistical Computing, Vienna, Austria). The data were analysed using descriptive statistics: means, SD, frequencies and percentages. No techniques were used to replace missing

data. Comparisons of clinical parameters between patients from the three phases were performed using the Kruskal-Wallis and $\chi^2$ tests. Elastic net regularisation was used to study the association between perceived limitations in daily life and perceived psychological burden with the following independent variables: age, gender, marital state (married or unmarried), education level (high, middle and low), pandemic phase, BDI, HCCQ, MoCA and TUG in seconds. Generally, elastic net regularisation leads to parsimonious models, which are easier to interpret.[11] Variable selection is performed by shrinking parameters towards zero and attenuating overfitting, a well-known problem if regression models are applied with a large number of predictors. Ten-fold cross-validation was applied to choose the best model with the lowest mean cross-validated error. Within the elastic net algorithm, variables remain in the model if the prediction error averaged over the ten cross-validation samples is reduced. In contrast to ordinary least squares regression or LASSO regularisation, the elastic net algorithm performs well in highly correlated variables, either including all of them with similar regression coefficients or excluding all of them from the best model. Regressions coefficients and p values of the model were reported. Elastic net regularisation was performed with the package glmnet in R V.3.6.2.

## RESULTS

Among the 452 people were interviewed in this study (table 1), 230 (50.9%) were interviewed in phase 1, 41 (9.1%) in phase 2 and 181 (40.0%) in phase 3. Only two people (0.4%) had COVID-19 infection and 18 people (4%) had been in quarantine before the interview. The people who were interviewed in the three phases did not differ in terms of age (p=0.68), BDI (p=0.16), HCCQ (p=0.76), MoCA (p=0.49), TUG (p=0.38), gender (p=0.53), marital state (p=0.58) and education level (p=0.69).

Overall, 307 (67.9%) people reported having relevant limitations in their daily routine. The perceived limitations in daily life caused by the measures decreased significantly during the pandemic phases (figure 1). The slight increase in perceived limitations from phase 2 to phase 3 was not significant. The reasons for the perceived limitations changed during the pandemic phases. At the beginning of the pandemic, people considered restricted social contacts and mobility as the most common reasons for perceived limitations in daily life. Later, at phase 3, wearing mouth–nose masks had become the main perceived limitation in daily life (figure 1). Corresponding to the easing of measures, social activities (meeting family members and friends) and visits to doctors increased from phase 1 to phase 3 (figure 2).

In the elastic net regularisation model higher perceived limitations in daily life were significantly associated with younger age and earlier pandemic phase (table 2). To a lesser extent, also other factors determined how strongly the limitations were felt in daily life. The restrictions were

**Table 1** Characteristics of the entire cohort

|  | M | SD |
|---|---|---|
| Age | 69.5 | 8.6 |
| Beck Depression Inventory II | 9.7 | 7.4 |
| Health Care Climate Questionnaire | 5.9 | 1.1 |
| Montreal Cognitive Assessment | 22.8 | 4.7 |
| Timed Up and Go | 10.6 | 4.5 |
|  | n | % |
| Sex |  |  |
| Female | 205 | 45.4 |
| Male | 247 | 54.6 |
| Marital status |  |  |
| Single | 131 | 29.5 |
| Married | 313 | 70.5 |
| Education level |  |  |
| High | 153 | 34.5 |
| Middle | 156 | 35.2 |
| Low | 134 | 30.2 |
| Diagnosis group |  |  |
| Movement disorders | 125 | 27.7 |
| Cerebrovascular disorders | 116 | 25.7 |
| Epilepsy | 24 | 5.3 |
| Neuromuscular disorders | 105 | 23.2 |
| Others | 82 | 18.1 |

perceived to be less drastic in daily life for married people, people with lower levels of education, and people with a poorer physician–patient relationship (lower HCCQ). In contrast, the limitations in daily life were perceived as more severe for men, people with a high level of education or higher MoCA, people with depression (BDI) and people with mobility impairments (higher TUG) (table 2).

Furthermore, the perceived psychological burden decreased during the pandemic phases (figure 3). Reason, such as fear of infection, worries on how it should go on (insecurity and concerns) and reduced social contacts, for the psychological burden did not remarkably change

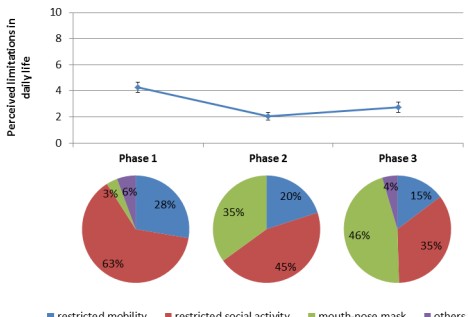

**Figure 1** Change and reasons of perceived limitations in daily life during the COVID-19 pandemic. 0='no limitation of my daily life' to 10='strongest restrictions of my daily life'

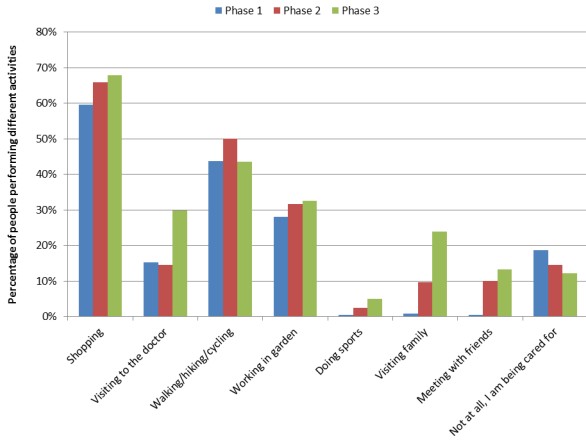

**Figure 2** Percentage of people performing different activities in different COVID-19 pandemic phases.

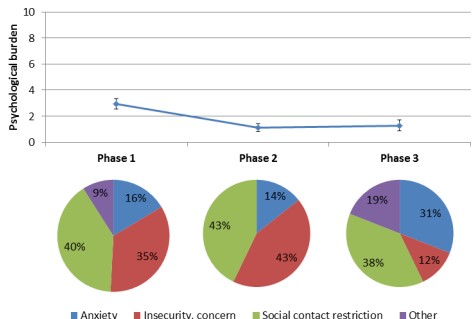

**Figure 3** Change and reasons of psychological burden during the COVID-19 pandemic. 0='not burdening at all' to 10='very strongly burdening'

during the phases (figure 3). In the elastic net regularisation model higher psychological burden was significantly associated with early pandemic phase, lower age and higher BDI. In addition, perceived psychological burden was higher in women, people with high education level or higher MoCA, people with mobility impairments, single living people and people with poorer physician–patient relationship (lower HCCQ) (table 3).

From 16 June 2020, the people were also asked if they approve the easing of measures. Among the 136 interviewed patients, 67 (49.3%) approved, 55 (40.4) did not approve and 14 could not make a decision. The common reasons not to approve the easing of measures were fear of increased risk of infection and irresponsible behaviour of other people. Patients who did or did not approve easing of measures did not differ in terms of age (p=0.85), gender (p=0.31), education level (p=0.80), marital state

(p=0.52), BDI (p=0.50), TUG time (p=0.27), MoCA (p=0.31) and patient–physician relationship according to the HCCQ (p=0.44). Both groups also did not differ in the perceived limitations in daily life (p=0.38) and psychological burden (p=0.51) caused by the measures.

## DISCUSSION

Our study shows how perceived limitations of daily life declined with ongoing easing of measures and the reasons for this change during different pandemic phases. Perceived limitations due to the measures were higher at the beginning of the pandemic and in younger aged persons. In our models, early pandemic phase and younger age were significant predictors of perceived limitations in daily life. There may be several reasons why older people perceived the limitations as less severe. For example, fewer responsibilities and more experiences with previous threatening situations can additionally explain the lower levels of perceived limitations among

### Table 2 Predictors of perceived limitations in daily living

|  | Coefficient | P value |
|---|---|---|
| Age * | −0.11 | <0.001 |
| Pandemic phase II * | −2.93 | <0.001 |
| Pandemic phase III * | −2.20 | <0.001 |
| Marital status (married) | −0.68 | 0.11 |
| Education (middle) | −0.80 | 0.08 |
| Education (low) | −0.49 | 0.31 |
| Health Care Climate Questionnaire | −0.13 | 0.47 |
| Male gender | 0.12 | 0.76 |
| Beck Depression Inventory II | 0.01 | 0.73 |
| Montreal Cognitive Assessment | 0.07 | 0.18 |
| Timed Up and Go | 0.03 | 0.47 |

$\chi^2(11) = 534.74$, p=0.00, Pseudo-$R^2$ (Cragg-Uhler)=0.23, Akaike-Information-Criterion (AIC)=1175.04, Bayesian-Information-Criterion (BIC)=1220.12.
*Significant predictors.

### Table 3 Predictors of perceived psychological burden

|  | Coefficient | P value |
|---|---|---|
| Age * | −0.06 | 0.01 |
| Pandemic phase II * | −2.62 | <0.001 |
| Pandemic phase III * | −2.07 | <0.001 |
| Marital status (married) | −0.63 | 0.12 |
| Education (middle) | −0.64 | 0.14 |
| Education (low) | −0.54 | 0.25 |
| Health Care Climate Questionnaire | -0.25 | 0.15 |
| Male gender | −0.55 | 0.13 |
| Beck Depression Inventory II * | 0.06 | 0.01 |
| Montreal Cognitive Assessment | 0.08 | 0.09 |
| Timed Up and Go | 0.02 | 0.62 |

$\chi^2(11)=599.29$, p=0.00, Pseudo-$R^2$ (Cragg-Uhler)=0.23, Akaike-Information-Criterion (AIC)=1301.61, Bayesian-Information-Criterion (BIC)=1348.00.
*Significant predictors.

the elderly.[12] To a lesser extent, factors other than the pandemic phase and age also played a role in how drastic the measures were experienced by the respondents. Independent of age and pandemic phase, the measures were particularly drastic in terms of daily life for men, educated people, people with depression and people with mobility impairments.

The most important limitation was the social contact restrictions. Even after various easing of the measures in phase 3, these were still the main reasons for perceived limitations in everyday life. Although perceived limitations were associated with younger age, the elderly people are particularly susceptible to the effects of uncertainty and lack of social contacts.[13] After the easing of the measures, the most frequent reason for perceived limitations in everyday life was the use of the mouth–nose mask (46%). This is in line with a recent cross-sectional Brazilian study of 1277 participants. Here, most respondents (67.3%) reported that the use of masks bothers them in some way (feel trapped, suffocated, feel shortness of breath, feel discomfort in the ears due to the elastics, fogs up the glasses).[14] Overall, even though people reported discontent over the mandatory use of masks, the general population followed the rule relatively well, as wearing masks signals empathy and prosocial behaviour toward members of the risk group.[15] Parallel to the easing of the measures, social activities, such as visiting friends also increased.

On the psychological burden side, a relevant fear of coronavirus infection is indicated throughout all pandemic phases. This fits the results of a representative survey of people in Wuhan (the epicentre of the COVID-19 outbreak in China) and Shanghai in February 2020. Here, the prevalence of moderate or severe anxiety was significantly higher in Wuhan than Shanghai. Anxiety was predicted with perceived harm of the disease and confusion about information reliability.[16] This fear increased with the significant easing of measures in phase 3. It is important to note here that the respondents were elderlies with various neurological and internal diseases. Many respondents stated that they were at increased risk of a severe course of COVID-19 disease and thus belonged to the risk group. This may explain the high level of anxiety.[8] Appropriately, approximately half of the respondents were critical on the easing of measures. However, the feeling of insecurity and worries about the future decreased over the course of the pandemic. This is the best indication that the respondents adapted to the specific requirements of the situation and were better informed than at the beginning of the pandemic, since sufficient information leads to less psychological burden like anxiety.[17] In the elastic net regularisation model the higher psychological burden was mainly associated with early pandemic phase, younger age and depression according to the BDI. While limitations in daily life were more relevant for men, higher psychological burden was associated with female gender. Moreover, high educated people, people with mobility impairments and single

living people claimed to be severely psychologically burdened by the measures.

The study has some limitations. It is important to emphasise that the data are cross-sectional, that is, different people were interviewed in the respective phases. Although the demographic and clinical parameters are comparable between the interviewed patients in the different phases, which is the usual design for other regular surveys in the context of the COVID-19 pandemic (eg, COSMO),[9] distortions caused by this design cannot be excluded. Furthermore, the survey focused on elderly and sick people. Although this limits the generalisability of the results, these data are important because older people are often under-represented in other surveys (eg, COSMO).

**Acknowledgements** We thank Marieke Jäger and Verena Buchholz for assistance in data acquisition.

**Contributors** AS, UT and HMZ collected data, performed the statistical analysis and drafted the initial manuscript. TP provided the concept and design of the study and was involved in the analysis. HMZ was involved in the interpretation of the data. TP revised the manuscript for important intellectual content. All authors read and approved the final manuscript.

**Funding** This work was supported by a Bundesministerium für Bildung und Forschung grant to Tino Prell (01GY1804).

**Competing interests** None declared.

**Patient and public involvement** Patients and/or the public were not involved in the design, or conduct, or reporting, or dissemination plans of this research.

**Patient consent for publication** Not required.

**Ethics approval** All procedures performed in studies involving human participants were in accordance with the ethical standards of the institutional research committee (ethics committee of the Jena University Hospital, 4572-10/15) and with the 1964 Declaration of Helsinki and its later amendments or comparable ethical standards. The study was approved by the local ethics committee (approval number 5290-10/17) of Jena University Hospital.

**Provenance and peer review** Not commissioned; externally peer reviewed.

**Data availability statement** Dataset is available on reasonable request from the corresponding author for scientific purpose.

**ORCID iD**
Tino Prell http://orcid.org/0000-0002-6423-3108

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
