## [Reviewer comments · BMJ Open]

ARTICLE DETAILS

TITLE (PROVISIONAL)	Changes of Perceptions and Behaviours during the Phases of COVID-19 Pandemic in German Elderly People with Neurological Disorders: An Observational Study using Telephone Interviews
AUTHORS	Zipprich, Hannah; Schönenberg, Aline; Teschner, Ulrike; Prell, Tino

VERSION 1 – REVIEW

REVIEWER	Donatella Rita Petretto Department of Education, Psychology, Philosophy University of Cagliari Italy
REVIEW RETURNED	16-Oct-2020

GENERAL COMMENTS	the paper is about a very interesting topic, but I suggest to better describe the relationship between clinical conditions of the elderly and their reply in the interview. There some typos in the text (example page seven, lines 12-17). I also suggest to better describe the results, with reference to other studies in the field (for example on the perception of the use of masks).
---

REVIEWER	Dr Kathryn Taylor University of Oxford UK
REVIEW RETURNED	20-Oct-2020

GENERAL COMMENTS	This study is about the effects of COVID-19 on an elderly population. The paper is well written in terms of structure and the standard of English is excellent. I have a number of comments and suggestions for improvement. 1. Abstract – It is not clear what Phase 3 means. The phases are defined later. The abstract should be stand-alone, so the definition for phase 3 should be included in the abstract. 2. Introduction – The study is about a specific group of the elderly (those with neurological problems), but the introduction, which should provide the motivation for the study, refers to the restrictions in detail and only refers briefly to elderly people in general. Those with neurological problems are not mentioned. 3. Reference list – it is very brief with only 11 references and only 4 in the introduction. This reflects the lack of justification for their study, which I do not doubt, but it needs to be more clear. 4. Inclusion criteria – The study population includes patients with cerebrovascular disorders and “other” (not specified). It is not clear that all the patients could give reliable assessments of their
---

	limitations. The mean MOCA score is high, but the range is not reported, and it seems that patients with dementia could be included in the population. 5. Statistical analysis – the 5% significance threshold for retaining variables in a regression model is standard. It is not clear what the 10% threshold is for. 6. Statistical analysis – The authors state that “Before regression analyses, This does not make sense, as the issues of possible autocorrelation and multicollinearity are considered within regression analyses. The authors should state that they “tested” for autoregression and multicollinearity, the methods they used, the outcomes, and how they dealt with what they found. For example, they do not report the Durbin-Watson statistic. 7. Statistical analysis – The authors seem to have tested for 2 assumptions of regression (autocorrelation and multicollinearity). Other assumptions also need to be tested (linear relationship, homoscedasticity, normality) 8. Figure 1 labelling is unclear – What exactly is plotted? Axis title is unclear (it refers to a score but could refer to a number of limitations). To remind the reader, a footnote should state what the score means i.e. 0 means no limitation etc 9. Figure 2 – The y axis needs a label 10. Figure 3 – The title needs to refer to a score and also a footnote added to remind the reader what the score means, as with Figure 1 11. The questionnaire responses at the end of the manuscript are unreadable in places due to the column widths being too narrow. I do not see what this adds.
--	--

VERSION 1 – AUTHOR RESPONSE

Reviewer: 1

COMMENT:

the paper is about a very interesting topic, but I suggest to better describe the relationship between clinical conditions of the elderly and their reply in the interview. There some typos in the text (example page seven, lines 12-17).

I also suggest to better describe the results, with reference to other studies in the field (for example on the perception of the use of masks).

ANSWER:

We thank the reviewer for this feedback. Yes, it is important to note in this study that people were interviewed who had various diseases and comorbidities. The relevant comorbidities (depression, cognitive disorder, limited mobility) were assessed by different scores. These were considered in the regression analyses. When entering the medical condition to these models, we found that the kind of neurological main diagnosis had no impact on the responses in the interview (i.e. perceived limitations in daily life and perceived psychological burden). In the revised version we used an elastic net regularization to explore the association between phases, clinical parameters and the perceived limitations in daily life / psychological burden. This supported our previous findings that perceived limitations in daily life are associated with the pandemic phase and age and that psychological burden is associated with pandemic phase, age, and depression. In addition in the models other clinical and demographical factors were found to be associated with perceived limitations in daily life and psychological burden. These are now reported in the revised version.

We tried to discuss the results in more detail in the context of current studies about COVID-19 pandemic, e.g. levels of anxiety in corona hotspots, bothering of masks. However, while many studies explored knowledge, attitudes and behaviors during the pandemic, little is known (in scientific literature) how the burden and limitations change over time.

Reviewer: 2

COMMENT: This study is about the effects of COVID-19 on an elderly population. The paper is well written in terms of structure and the standard of English is excellent. I have a number of comments and suggestions for improvement.

COMMENT: 1. Abstract – It is not clear what Phase 3 means. The phases are defined later. The abstract should be stand-alone, so the definition for phase 3 should be included in the abstract.
ANSWER: Thank you for this advice. We revised this for clarity.

COMMENT: 2. Introduction – The study is about a specific group of the elderly (those with neurological problems), but the introduction, which should provide the motivation for the study, refers to the restrictions in detail and only refers briefly to elderly people in general. Those with neurological problems are not mentioned.
ANSWER: Thank you for this advice. We revised accordingly. In fact, many surveys on corona pandemic have not really taken older people into account. Therefore we lack data especially for this cohort. Furthermore, little is known about how the measures are perceived by older people with chronic diseases or relevant functional impairments. In our opinion, the results of surveys of younger or community-dwelling elderly people, e.g. COVID-19 Snapshot Monitoring (COSMO) 9), are not necessarily transferable to this cohort. Therefore, we interviewed elderly people who were previously hospitalized in our hospital because of a neurological disorder. We were particularly interested in how the perception of the measures changes over time.

COMMENT: 3. Reference list – it is very brief with only 11 references and only 4 in the introduction. This reflects the lack of justification for their study, which I do not doubt, but it needs to be more clear.
ANSWER: Thank you for pointing this out. We added more literature in the revised manuscript. There are many studies analyzing aspects of mental health during COVID-19 pandemic; but of course we can only cite a few for this short report. However, these studies are mainly surveys in younger (e.g. students, people using social media etc) people. In particular, no study so far addressed changes of perception during the phases in elderly people. Moreover, little is known how easing of measures influences mental health.

COMMENT: 4. Inclusion criteria – The study population includes patients with cerebrovascular disorders and “other” (not specified). It is not clear that all the patients could give reliable assessments of their limitations. The mean MOCA score is high, but the range is not reported, and it seems that patients with dementia could be included in the population.
ANSWER: Thank you for making this relevant advice. Indeed persons with mild cognitive deficits were not excluded. Given the high prevalence of cognitive deficits in the elderly population this would dramatically decrease generalizability. Nevertheless, we made sure that no patients were included who could not validly answer questionnaires or interview questions. The baseline data in the NeuGerAdh study was collected by trained research staff. Using the Montreal Cognitive Assessment (MOCA), we assessed the cognitive status after a short introduction to the aims and methods of the study. A valid impression of each patient’s ability to understand and complete a questionnaire was achieved with face-to-face testing. Thus, patients with a MOCA score below common thresholds of dementia were included if they could understand and answer the questionnaires coherently (e.g. lower score when fine motor skills were impaired). Then, we collected the clinical data. The telephone

interview was performed 12 months after the hospital stay. We do not assume that a relevant proportion of patients developed frank dementia in that time period. Moreover, also the telephone interviews were collected by trained research staff to ensure high data quality and valid data.

COMMENT: 5. Statistical analysis – the 5% significance threshold for retaining variables in a regression model is standard. It is not clear what the 10% threshold is for.

ANSWER: Stepwise methods include or remove one independent variable at each step, based on the probability of p-value. The limits for the criteria controlling variable inclusion or removal can be specified by defining probabilities. In our backward selection model, first all variables were entered into the equation and then sequentially removed. At each step, the largest probability of F is removed (if the value is larger than 0.10). In the revised version, we replaced the traditional regression model by the more up-to-date elastic net regularization. Please see the answer to your comment 6.

COMMENT: 6. Statistical analysis – The authors state that “Before regression analyses, This does not make sense, as the issues of possible autocorrelation and multicollinearity are considered within regression analyses. The authors should state that they “tested” for autoregression and multicollinearity, the methods they used, the outcomes, and how they dealt with what they found. For example, they do not report the Durbin-Watson statistic.

ANSWER: Thank you for this feedback. Of course testing for autocorrelation and multicollinearity is part of the analysis itself and our wording is confusing. With regard to Durbin-Watson the model for perceived psychological burden showed adequate Durbin Watson = 1.8. However, in the model for perceived limitations in daily life the Durbin Watson was lower (1.3). In the revised version, we therefore performed analyses using an elastic net regularization. Generally, elastic net regularization leads to parsimonious models, which are easier to interpret. Variable selection is performed by shrinking parameters towards zero and attenuating overfitting, a well-known problem if regression models are applied with a large number of predictors. Ten-fold cross validation was applied to choose the best model with the lowest mean cross-validated error. Within the elastic net algorithm, variables remain in the model if the prediction error averaged over the ten cross-validation samples is reduced. In contrast to ordinary least squares regression or LASSO regularization, the elastic net algorithm performs well in highly correlated variables, either including all of them with similar regression coefficients or excluding all of them from the best model. Regression coefficients of the model with 95% confidence intervals (CI) were reported. Elastic net regularization was performed with the package glmnet in R 3.6.2 (R Foundation for Statistical Computing, Vienna, Austria).

The elastic net regularization models supported in principle our previous findings: Perceived limitations in daily living were associated with pandemic phase and age. Perceived psychological burden was associated with pandemic phase, age and depression. In order to avoid confusion and to provide state of the art statistical methods we decided to report only the findings from the elastic net models. This has also made it possible to better understand which clinical and demographic factors are related to perceived limitations in daily life and perceived psychological burden.

COMMENT: 7. Statistical analysis – The authors seem to have tested for 2 assumptions of regression (autocorrelation and multicollinearity). Other assumptions also need to be tested (linear relationship, homoscedasticity, normality)

ANSWER: Thank you for pointing out this omission. Of course we checked for all assumptions. Given that multicollinearity is a common problem in many studies using regression analyses, we mentioned this separately. As detailed in above, we revised this section.

COMMENT: 8. Figure 1 labelling is unclear – What exactly is plotted? Axis title is unclear (it refers to a score but could refer to a number of limitations). To remind the reader, a footnote should state what the score means i.e. 0 means no limitation etc

ANSWER: Thank you for this advice. We added a footnote to the figure legend.

COMMENT: 9. Figure 2 – The y axis needs a label
ANSWER: Thank you for this advice. We added a axis title.

COMMENT: 10. Figure 3 – The title needs to refer to a score and also a footnote added to remind the reader what the score means, as with Figure 1
ANSWER: Also here we added a footnote to the figure legend.

COMMENT: 11. The questionnaire responses at the end of the manuscript are unreadable in places due to the column widths being too narrow. I do not see what this adds.
ANSWER: As recommended by the editorial office, we deleted this section.

We appreciate the suggestion and hope that we have done this to the Reviewer's satisfaction in the revised version.

VERSION 2 – REVIEW

REVIEWER	Donatella Rita Petretto University of Cagliari
REVIEW RETURNED	12-Dec-2020

GENERAL COMMENTS	The authors addressed all issues.
-----------------------------------

REVIEWER	Dr Kathryn Taylor University of Oxford UK
REVIEW RETURNED	14-Dec-2020

GENERAL COMMENTS	I am happy with the changes to the manuscript in response to my comments.
---